# Review of Interface Passivation of Perovskite Layer

**DOI:** 10.3390/nano11030775

**Published:** 2021-03-18

**Authors:** Yinghui Wu, Dong Wang, Jinyuan Liu, Houzhi Cai

**Affiliations:** Key Laboratory of Optoelectronic Devices and Systems of Ministry of Education and Guangdong Province, College of Physics and Optoelectronic Engineering, Shenzhen University, Shenzhen 518060, China; yinghui@szu.edu.cn (Y.W.); wangdong20@szu.edu.cn (D.W.); 20142702022@cqu.edu.cn (J.L.)

**Keywords:** interface passivation, perovskite solar cells, organic materials, inorganic materials

## Abstract

Perovskite solar cells (PSCs) are the most promising substitute for silicon-based solar cells. However, their power conversion efficiency and stability must be improved. The recombination probability of the photogenerated carriers at each interface in a PSC is much greater than that of the bulk phase. The interface of a perovskite polycrystalline film is considered to be a defect-rich area, which is the main factor limiting the efficiency of a PSC. This review introduces and summarizes practical interface engineering techniques for improving the efficiency and stability of organic–inorganic lead halide PSCs. First, the effect of defects at the interface of the PSCs, the energy level alignment, and the chemical reactions on the efficiency of a PSC are summarized. Subsequently, the latest developments pertaining to a modification of the perovskite layers with different materials are discussed. Finally, the prospect of achieving an efficient PSC with long-term stability through the use of interface engineering is presented.

## 1. Introduction

Organic–inorganic hybrid perovskite materials exhibit excellent photoelectric properties (e.g., an adjustable band gap, a long carrier life, a long carrier diffusion length, a low exciton binding energy, a high molar extinction coefficient, low cost, and an easy preparation) and are widely used in various optoelectronic devices, such as solar cells, light-emitting diodes, photodetectors, memristors, and sensors [1,2,3,4,5,6,7,8]. Among such devices, perovskite solar cells (PSCs) have been the most frequently investigated [9]. According to calculations based on the Shockley–Queisser limit theory, a light-absorbing material with a bandgap of 1.6 eV can achieve a theoretical limit efficiency of approximately 30.5% [10]. Current single-junction PSCs have achieved a certified power conversion efficiency (PCE) of 25.5% [11,12]. However, this value is still lower than the theoretical efficiency; therefore, there remains room for improving the performance of PSCs.

In a PSC, the most commonly used perovskite material is the organic–inorganic metal halide perovskite, CH_3_NH_3_PbX_3_ (MAPbX_3_), the structure of which is shown in Figure 1. Its disadvantages include a poor stability and low open-circuit voltage. Therefore, different metal cations, organic cations, and halogens can be used to replace the corresponding elements or ions in traditional materials [13]. In addition, HC(NH_2_)_2_(FA) and Cs can be used to replace MA owing to their wider absorption range; consequently, a bandgap corresponding to the entire wavelength range of visible light can be achieved, thereby resulting in a higher energy conversion rate [14,15]. Tin-based PSCs exhibit electronic structures and semiconductor properties that are similar to those of perovskites, including a high absorption coefficient, high carrier mobility, and an ideal bandgap [16,17,18]. This is ideal for the manufacture of environmentally friendly PSCs. As promising alternatives to tin, transition metals such as Fe, Cu, Mn, Ni, and Co are rich in reserves and contain diverse ion valences [19]. The addition of halogen elements Cl and Br promotes a non-uniform nucleation at different positions and improves the morphology of perovskite films, thereby facilitating carrier transport and reducing the carrier recombination [20,21,22]. Improvements in the morphology and microstructure of the film by optimizing the preparation process for control of the nucleation density and growth rate of the crystal can increase the PCE. A one- (Figure 2a) or two-step (Figure 2b) method is typically used to prepare high-performance PSCs [23,24,25]. Furthermore, the dual-source vapor deposition method, two-step continuous solution-vapor assisted deposition method, and other approaches have been applied to the optimization of perovskite films for improving the PCE of PSCs [15,26,27].

In addition to the development of new materials and the optimization of the preparation process, the selection of interface materials and the generation, collection, and transport efficiency of interface charges are crucial to the performance of a PSC [2,3,4,14]. The typically investigated flat-panel sandwich-structured PSCs are disadvantageous owing to their significant interface recombination, low carrier separation efficiency, and low stability, all of which have restricted their development. Many studies have proven that the efficiency and stability of perovskite solar cells are closely associated with the nonradiative recombination loss of bulk and interface carriers [28]. Effective methods for the further development of PSCs include optimizing their structure, improving the charge separation and extraction rate at the interface, and reducing the effect of the organic material degradation on the device stability [15,29]. The interface engineering of PSCs is discussed herein, primarily in terms of the effect of the light-absorbing layer on the interface and interface materials that can be inserted into the perovskite layer. In addition, recent important breakthroughs in interface engineering and possible future developments are summarized.

## 2. PSCs

### Structure and Principle of PSC

A PSC comprises a positive electrode layer, a hole/electron transport layer, a light absorption layer, an electron/hole transport layer, and a negative electrode layer (Figure 3a) [1,13,30,31]. The transport layer transports electrons and holes while preventing them from entering.

Perovskite materials have a high dielectric constant and low excitation energy. After absorbing sunlight, the perovskite layer first absorbs photons to generate electron–hole pairs [32,33]. Owing to the difference in the binding energy of excitons in perovskite materials, these carriers either become free carriers or form excitons. (Figure 3b). These perovskite materials tend to have a low carrier recombination probability and high carrier mobility; therefore, the diffusion distance and lifetime of the carrier are long. These uncomplexed electrons and holes are collected by the electron transport layer (ETL) and hole transport layer (HTL), respectively, i.e., the electrons are transported from the perovskite layer to the isoelectron transport layer and finally collected by the fluorine-doped tin oxide (FTO). The holes are transported from the perovskite layer to the hole transport layer and are finally collected by the metal electrode.

These processes are accompanied by some carrier losses, such as a reversible recombination of electrons and holes in the ETL and perovskite layer, the recombination of electrons and holes in the ETL and HTL, and the recombination of electrons and holes in the perovskite layer and HTL, respectively. The loss of these carriers should be minimized to improve the overall performance of the device. Finally, a photocurrent is generated through a circuit connecting the FTO and metal electrode [31,34,35]. One of the key methods for improving the PCE of the PSCs is by improving the carrier separation efficiency of the materials.

## 3. PSC Interface

Several important interfaces exist in a PSC, such as the perovskite/ETL interface and the perovskite/HTL interface. In addition, several important physical processes occur at these interfaces, such as band bending, carrier injection, carrier recombination, charge accumulation, and ion migration. Interface defects contribute significantly to the nonradiative recombination of the interface. The interface between the transport and perovskite layers determines whether the photogenerated carriers can be smoothly transferred from the light-absorption layer to the n/p-type semiconductor [13,29,36,37]. The recombination of charges at the interface severely restricts the photovoltaic performance of PSCs.

### 3.1. Interfacial Band Arrangement of PSCs

The problem of energy level matching between the interface material and the perovskite is essential for the development of a better PSC. Electrons and holes pass through each layer and encounter many different interfaces (Figure 4a,b) (e.g., the electrons will coincide with the perovskite layer/ETL interface and ETL/negative electrode interface, and the holes will coincide with the perovskite layer/HTL interface and HTL/positive electrode interface) [13]. Whether the electrons and holes can pass through those interfaces smoothly depends significantly on whether the energy levels at the interfaces match. The matching of the interface energy level directly affects the collection efficiency of the carriers and hence the final PCE of the device [38,39]. An imperfect interface energy band arrangement will cause a non-radiative recombination of the interface. Regardless of whether the device has a porous/planar, n–i–p, or p–i–n structure, an optimization of multiple interfaces is required [1,4,15,40]. A perfect interface band arrangement is not only conducive to an efficient charge extraction, transmission, and collection, but is also conducive to reducing the interface charge accumulation and J–V hysteresis [41,42].

When materials of different functional layers are in contact with each other to form an interface, energy level bending and a charge transfer may occur, causing the vacuum energy level to no longer be leveled, which affects the energy level matching at the interface. The energy level arrangement is affected by several factors, such as the energy level position of the adjacent interface material, interface defects, ion migration, interface instability, and preparation conditions of the adjacent interface layer [10,29,43]. The introduction of the interface materials into the perovskite interface can significantly affect the interface band arrangement and interface carrier dynamics, thereby increasing the open-circuit voltage or effective carrier injection. The movement direction of the material work function depends mainly on the electron cloud density, dipole moment, space concept, and molecular structure of the interface material [28,44]. Various interface materials have been developed to modify the interface in the PSCs, the construction of gradient wide band gap perovskite components, quantum dots, self-assembled monolayers, wide band gap insulating materials, charge transport materials, and small organic molecules [28,45,46,47,48].

### 3.2. Defects at Interface

Similar to other solar cells, the performance of a PSC is governed by its interface. To investigate a PSC, one must understand and rationally regulate their physical processes. Therefore, similar to a bulk nonradiative recombination, minimizing the interface nonradiative recombination loss is crucial for achieving efficient and stable PSCs without hysteresis [10,49]. It has been widely reported that the efficiency, stability, and hysteresis effects of PSCs are closely associated with the nonradiative recombination dynamics of the interface carriers. The main causes of an interface nonradiative recombination loss are the interface defects, imperfect energy level arrangements, and interface reactions. The bulk defects of most perovskite films are shallow point defects, whereas most of the defects at the interface are deep-level and high-latitude defects (such as surface defects, grain boundary defects, and precipitated phase defects) [28,29,50,51,52].

The perovskite film prepared using a traditional low-temperature solution method is typically a polycrystalline film, which is prone to pinholes and defects on the grain boundary or surface [53,54,55]. The defects generated can not only capture photogenerated carriers, limit the diffusion of the carriers, and reduce the lifetime of the carriers, they also cause ion migration and diffusion, ultimately resulting in a decrease in the device stability and efficiency. In this study, the defect density at the interface of a polycrystalline film was one to two orders of magnitude higher than that of the defects inside the film. The interfacial deep-level defects can capture carriers at the heterojunction interface and cause nonradiative recombination losses at the interface, thereby reducing the cell efficiency and stability [42,56]. Carriers trapped by defects and charged ions that migrate from the bulk phase to the surface and grain boundaries will accumulate at the interface, resulting in band bending, changes in the energy level arrangement and built-in electric field, and nonradiative recombination losses at the interface, none of which are conducive to a carrier separation, injection, or transmission [57,58].

### 3.3. Chemical Reaction at Interface

In addition to the band arrangement and defects, the chemical reaction has been proven to cause a nonradiative recombination at the interface [10,59]. An interface reaction is caused either by a direct chemical reaction between the interface materials or the reaction between the ions in the perovskite layer and the charge transport layer or electrode. An interface reaction induced by contact can inhibit ion migration or diffusion by introducing chemically inert functional molecules [60]. Furthermore, it can prevent the occurrence of interfacial chemical reactions. The charge recombination loss in the grains of perovskite films prepared using the current technology is almost negligible. However, several studies have shown that the defect state density of polycrystalline perovskite films is several orders of magnitude higher than that of single-crystal perovskite films. This is primarily because of unavoidable structural defects such as uncoordinated Pb^2+^ at the surface of the polycrystalline perovskite film [61,62]. These defect sites not only capture photogenerated carriers and cause energy loss, they also easily adsorb substances such as moisture and oxygen in the environment, thereby accelerating the degradation of the perovskite film.

Vacancy site defects can result in a much faster ion migration at the grain boundary and surface compared with that in the bulk [63,64,65,66]. Eliminating ion migration paths by passivating bulk and interface defects can effectively inhibit ion migration or diffusion. The interface reaction effectively eliminates the surface defects of semiconductor materials and improves the photoelectric performance of the device. Organic molecules contain “Lewis base” functional groups, such as thiophene, pyridine, mercaptans, and compounds containing carbonyl or cyano groups [67,68,69]. They can coordinate with Pb^2+^ on the surface of a perovskite film through a lone pair of electrons, thereby passivating the perovskite film. The interaction between the surface of a perovskite film and the functional groups of the passivation molecule can further passivate the grain boundary, including those that are deep in the substrate. This is an effective method for preventing interface chemical reactions by introducing a suitable layer at the interface to physically separate two interface materials in close contact with each other.

## 4. Interface Modification

Defects in the perovskite absorber layer and interface defects are internal factors that affect the stability of PSCs. When PSCs are operated under light or a bias voltage, their defect distribution becomes unstable. In addition, these defects can trap photogenerated charges to form a local electric field, drive ions to redistribute, and cause a phase separation, thereby reducing the stability and photoelectric conversion performance of the PSCs. The passivation of interface defects can effectively yield efficient and stable PSCs. Various interface materials have been developed to modify the interface of a PSC [49,51,70]. The interface materials that have been reported thus far can be classified into two categories: inorganic and organic.

### 4.1. Organic Material

Tightly arranged organic interface materials formed by strong physical forces can be used to obtain efficient and stable PSCs.

#### 4.1.1. Organic Salt

A perovskite film prepared using a general solution method contains a large number of grain boundaries and defects such as dislocations, impurities, and vacancies owing to the fracture of chemical bonds that are easily formed at the grain boundaries [50,71,72,73,74]. The passivation of the grain boundary defects can effectively improve the efficiency and stability of PSCs [75,76,77]. Therefore, identifying a suitable passivation agent is essential for preparing efficient and stable PSCs. Researchers have discovered that organic amine salts can be used to passivate the surface of a perovskite film and increase the open-circuit voltage of the device to be fabricated. A brief summary is given in Table 1.

Essa et al. used different types of ammonium salts (i.e., ethylammonium, imidazolium, and guanidine iodide) to treat the surface of a perovskite film to reduce electronic defects at the interface between the perovskite film and HTL. This treatment increased the power conversion efficiency from 20.5% in the control group to 22.3%, 22.1%, and 21.0% for devices treated with ethylammonium, imidazolium, and guanidine iodide, respectively. The device with the best performance indicated a maximum power tracking of 550 h, and the efficiency loss under a full sunlight intensity was only 5% [78]. With the use of phenethylammonium iodide (PEAI) on FAMA hybrid perovskite films for surface defect passivation to increase ***V_oc_*** of the device, the ***V_oc_*** reached 1.18 V. PEAI can be formed on the surface of perovskite and produce more efficient PSCs by reducing defects and inhibiting the nonradiative recombination. Finally, planar PSCs with a certified efficiency of 23.32% were obtained [79]. Hongwei Zhu et al. displayed that a simple surface treatment using 4-tert-butyl-benzylammonium iodide (tBBAI) can significantly accelerate the charge extraction from a perovskite to the spiro-OMeTAD hole-transport agent, while hindering the non-recombination of radiated charge carriers. The large tert-butyl group of tBBAI prevents harmful aggregation owing to a steric repulsion. This enables the PCE of the PSCs to increase from approximately 20% to 23.5%, thereby reducing the hysteresis to an almost undetectable level [56]. The use of organic salts is considered to be an efficient device architecture for PSCs. By introducing the in situ reaction of n-hexyltrimethylammonium bromide (HTAB) on the surface of perovskite, a thin wide-bandgap perovskite layer was formed on top of a light absorption layer with a narrow bandgap. HTAB prevents electron transfer between the perovskite layer and poly (3-hexylthiophene) (P3HT); it effectively reduces the charge recombination at the interface. Based on the P3HT without any doping, the certified efficiency of the device was discovered to be 22.7%. [80].

#### 4.1.2. Graded Perovskite

While the efficiency of the PSCs has gradually progressed, the stability of perovskites to light, heat, water, and oxygen is a challenge for industrial applications. Constructing a perovskite structure with a quasi-two-dimensional (2D) layered structure has proven to be an effective strategy for improving the environmental stability of PSCs [81,82]. A small number of ammonium salts participate in the ion exchange reaction, and 2D perovskite is formed on the surface of a three-dimensional (3D) perovskite film [83,84,85]. Therefore, the introduction of a small amount of 2D perovskite as the passivation layer for a 3D perovskite film results in the following: First, it reduces the grain boundary and surface ion defects. Second, the vacuum energy level at the interface is bent, thereby improving the band matching of the interface. Finally, transport of the holes is prevented [86,87]. The graded perovskite systems are also summarized in Table 1.

The introduction of hydrophobic fluorinated aromatic amine large-volume cation strategy on the surface of the 3D perovskite films forms a 3D-2Dstacked perovskites, which combines the stability of 2D perovskite and the high performance of 3D perovskite to improve the efficiency and stability of the devices [87]. Zhou et al. systematically compared the effects of 3D–2D stacked perovskites on the performance and stability of a device by treating the surface of the 3D perovskite with aromatic cations placed at different positions [88]. The hydrophobic structure not only blocks the invasion of water molecules, but also can passivate the surface defects of the perovskite and effectively promote the transport of holes to the HTM layer. A 1-naphthylmethylammonium bromide dynamic spin-coating method to generate a 2D/3D perovskite heterojunction in situ was used. This heterojunction facilitates the separation of electrons and holes and improves the efficiency of the PSCs. The maximum PCE yielded by the PSCs was 21.09%. In addition, a hydrophobic 2D ultrathin perovskite layer can improve the humidity stability of the PSCs. It was reported that by using this method, a PSC maintained an initial efficiency of 80% for 350 h at 70–80% relative humidity [89]. Meanwhile, Wang et al. utilized the amphiphilic structure of large molecule S-benzyl-L-cysteine and its unique solubility behavior. In another study, a 2D/3D heterojunction MAPbI_3_ perovskite was prepared based on a self-assembly one-step knife coating method. The 2D/3D series module (active area of 10.08 cm^2^) achieved a maximum efficiency of 15.38%, and the average open-circuit voltage of the sub-cell showed a maximum value of 1.17 V [90].
nanomaterials-11-00775-t001_Table 1Table 1Performance summary of organic salt passivation and graded PSCs.PassivationMaterialsPerovskite*J_sc_*(mA·cm^−2^)*V_oc_* (V)*FF* (%)*PCE* (%)YearRef.tBBAICs_0.05_FA_0.85_MA_0.10_Pb(I_0.97_Br_0.03_)_3_25.101.14282.123.52020[56]PEAICs_0.05_FA_0.85_MA_0.10_Pb(I_0.97_Br_0.03_)_3_25.011.12280.922.72020[56]EAIFA_0.9_Cs_0.07_MA_0.03_Pb(I_0.92_Br_0.08_)_3_24.361.12380.422.302019[78]IAIFA_0.9_Cs_0.07_MA_0.03_Pb(I_0.92_Br_0.08_)_3_24.141.10379.421.602019[78]GuaIFA_0.9_Cs_0.07_MA_0.03_Pb(I_0.92_Br_0.08_)_3_24.451.10675.320.92019[78]PEAI(FAPbI3)_1__−x_(MAPbBr_3_)_x_25.21.18078.423.22019[79]HTAB(FAPbI3)_0.95_(MAPbBr_3_)_0.05_24.881.15281.423.32019[80]NMAICs_0.05_(MA0.15FA_0.85_)_0.95_Pb(I_0.85_Br_0.15_)_3_22.281.17477.6320.572020[87]oFPEAICs_0.05_FA_0.79_MA_0.16_PbI_2.49_Br_0.51_22.621.1777.920.602020[88]mFPEAICs_0.05_FA_0.79_MA_0.16_PbI_2.49_Br_0.51_22.751.17576.820.522020[88]pFPEAICs_0.05_FA_0.79_MA_0.16_PbI_2.49_Br_0.51_22.231.15379.520.372020[88]NMABrCsFAMAPb(I)_3_23.621.1379.021.092020[89]SBLCMAPbI_3_22.361.1975.720.142020[90]C_4_Br(FAPbI_3_)_0.92_(MAPbBr_3_)_0.08_24.51.158123.22019[91]C_6_Br(FAPbI_3_)_0.92_(MAPbBr_3_)_0.08_24.61.1681.423.22019[91]C_8_Br(FAPbI_3_)_0.92_(MAPbBr_3_)_0.08_24.51.1682.023.32019[91]PEAIFASnI_3_19.450.3770.104.892018[92]PZPYCs_0.04_MA_0.16_FA_0.8_PbI_0.85_Br_0.15_21.701.0877.018.102018[93]PAICsFAMAPb(IBr)_3_23.691.1180.7621.192020[94]CsPbI_2_Brnanosheet–CsPbI_2_Br quantum dotsCsPbBrI_2_12.931.190.8012.392018[95]

The passivation material significantly reduces the defects and non-radiative recombination between the perovskite active layer and the transport layer. Therefore, the transfer of electrons from the perovskite layer to the anode is greatly increased. Yang et al. discovered that unlike a traditional p/n-type heterojunction formed through pure 2D/3D perovskites (n = 1, ∞), the C_61_-butyric acid methyl ester (PCBM) interface, and quasi-2D perovskites (n = 3, 5, 40), a junction was formed when the PCBM interface was involved, thereby reducing the hole concentration at the interface and inhibiting an interface recombination [96]. Yoo, Jason J et.al displayed a (linear alkyl ammonium bromide/chloroform) post-processing strategy that can effectively passivate the interface and grain boundary defects and increase the moisture resistance to generate layered perovskite in situ on a 3D perov-skite film. This results in an open-circuit voltage loss of only ~340 mV, a maximum efficiency of 23.4%, a certification efficiency of 22.6%, and a significantly improved stability. In addition, the PSCs achieve a maximum electroluminescence EQE of 8.9% [91]. Furthermore, N-butylammonium, N-(3-aminopropyl)-2-pyrrolidinone, and 5-ammoniumvaleric acid iodide have been used to realize 2D/3D perovskite heterostructures. Similarly, the introduction of large organic cations in tin perovskites, including n-butylammonium, phenethylammonium, and guanidine, can yield low-dimensional wide-bandgap tin perovskites, effectively stabilizing Sn^2+^ and reducing the energy level mismatch of tin PSCs [1,92,97].

In addition to 2D materials, zero-dimensional (0D) and one-dimensional (1D) materials have been used to investigate passivation perovskite layers [93,94]. Long-term stability is a basic requirement for the commercialization of PSCs. The heterojunction constructed by low-dimensional and three-dimensional perovskite (1D/3D) can improve the stability of PSC. An in situ cross-linked polymerizable propargyl ammonium to a 3D perovskite film at the surface and grain boundaries to form a 1D/3D perovskite heterostructure was introduced. This passivation strategy not only significantly improves the transport of the interface carrier, it also releases the residual tensile strain in the perovskite film. After 3000 h of continuous illumination under the maximum power point (MPP) operating conditions, the corresponding device achieved an optimal PCE of 21.19% [94]. The introduction of 1D materials at the 1D–3D perovskite heterogeneous interface reduces the electron and hole defect state density of the film. Simultaneously, the increase in the depletion region accelerates the transport of carriers and reduces the recombination of carriers [93]. This conclusion was supported by the work of Fan et al. CsPbBrI_2_ PSCs using CsPbBrI_2_ distributed in a bulk-nanosheet-quantum dot or 3D–2D–0D size profile interface structure was constructed. Consequently, a heterojunction with a gradual size was formed. Hence, the energy was optimally arranged between the valence and conduction bands of the CsPbBrI_2_ layer and HTL [95]. These studies have effectively improved the thermal stability of the device and provided new ideas and methods for the preparation of efficient and stable PSCs.

#### 4.1.3. Chemical Passivation

Many passivation molecules directly neutralize the surface charges or dangling bonds to annihilate the corresponding electron traps [98,99,100], e.g., by introducing Lewis bases (such as thiourea) to coordinate with Pb^2+^ on the surface and grain boundaries to reduce the defect density; some electron trap states can accept one of the Lewis-based defects on the surface of the perovskite through the Lewis acid molecule electron to be reduced [62,101,102,103,104]. The molecular hydrophilic chemical group and perovskite molecule combine to form a coordination bond, which is conducive to the transport of carriers [60,105,106]. Furthermore, the introduction of molecules enhances the smoothness of the perovskite layer interface, increases the contact angle, reduces pores and defect states, and significantly improves the interface properties of the perovskite layer [98,107]. In addition, the energy level matching between the perovskite layer and the carrier transport layer is altered, and the light absorption range of the material is increased, thereby resulting in a higher *Jsc* and *FF* [108,109]. The results of selected studies are summarized below (as listed in Table 2).

The structure of passivating molecular functional groups, including carboxyl, amino, isopropyl, phenethyl, and tert-butyl phenethyl groups was systematically designed, and their passivation ability in perovskites could be investigated. It was discovered that carboxyl and amine groups can remedy the charging defects through electrostatic interactions, and that the aromatic structure can reduce neutral iodine-related defects. The interaction between the perovskite surface and the molecules can result in grain boundary passivation, including those deep in the substrate. Hence, a new passivation molecule, D-4-tert-butylphenylalanine, was designed to create a high-performance *p*–*i*–*n* solar cell with a stable efficiency of 21.4% [61]. The passivation molecular structure not only reduced the open circuit voltage loss of the PSCs, but also improved the stability of the device. An effective TBPO–perovskite Coulomb interaction and intermolecular π–π conjugation was utilized to stably passivate the perovskite surface, thereby realizing efficient and stable PSCs. TBPO can significantly inhibit deep defects in perovskites and reduce the charge trapping cross-section by two orders of magnitude. TBPO molecules realize superstructures through a π–π conjugated self-assembly, thereby improving the passivation stability. After 250 h of continuous light, the MPP maintained an initial efficiency of 92%, which significantly improves the stability of the PSCs, particularly when the initial aging phenomenon is eliminated [109]. A series of D–π–A porphyrin molecules were developed and used to treat the surface of the perovskites. The cyanoacrylic functional group of porphyrin molecules successfully passivates defects on the surface and the grain boundary of the perovskite lattice, the former of which results in an inhibition of nonradiative recombination be-tween the perovskite/HTL interface. The existence of long alkyl chains prevents the penetration of water molecules into the air. The passivated device achieved a maximum efficiency of 22.37% [98].

A hydrophobic conjugated polymer with high mobility and multiple passivation functional groups was also used as the interface passivation layer to improve the humidity stability of the perovskite film. The quality and hydrophobicity of the perovskite film improved significantly through DPP-DTT as the interface modification layer, whereas the PCE of the prepared CsPbI_2_Br PSCs reached 15.14%. After 22 d in air, its PCE remained at above 95% of its initial efficiency and it managed to withstand more than 25 h of continuous thermal stress at 85 °C [110]. Although some materials do not form a chemical reaction with perovskite materials, PCBM and PMMA have been successfully used to passivate defects in perovskites [111,112,114,115]. Small-sized passivation molecules can be spontaneously distributed in the grain boundary to enhance the passivation effect of the grain boundary defects of hybrid perovskites [48,58,100,116,117].

### 4.2. Inorganic Material

The attenuation of perovskites is due to the defect sites on the surface and grain boundaries, which are more sensitive to water and oxygen. Passivation treatment, which uses hydrophobic organic molecules or polymer materials to physically shield the surface defects of a perovskite material, has achieved a certain effect. The bond between the shielding molecules and perovskite molecules is insufficient to protect the material from water and oxygen for a lengthy period of time. Inorganic materials have a large bandgap that can reduce the surface recombination speed and change the surface work function energy level for a better match [118,119,120,121]. They are chemically stable in humid air and can be used as a passivation material for lead–halide perovskites. The reported performances of some high-efficiency PSCs are summarized in Table 3.

#### 4.2.1. Quantum Dots

The large specific surface area, rich surface functional groups, good electrical conductivity, adjustable work function, unique nanosize effects, and surface effects of quantum dots render them suitable for use in solar cells [122,137,138]. Quantum dots effectively passivated perovskite crystal defects and promoted the stability of battery performance. The oleylamine-coated PbSO_4_(PbO)_4_ quantum dots were reported by Chen et al., which are dual-function blunt chemical materials, can simultaneously passivate surface defects, and prevent moisture and oxygen from penetrating into the perovskite layer, thereby realizing stable and efficient PSCs. PbSO_4_(PbO)_4_ quantum dots reduce the defect density of the perovskite film by passivating insufficiently coordinated Pb and I anions. In addition, the hydrogen bond between the H atom of OA and the I atom of the perovskite, as well as the interfacial electric field at the perovskite/OA interface, can improve the efficiency and stability of the PSC. In their study, a PSC comprising PbSO_4_(PbO)_4_ with OA coating achieved an efficiency of 20.02%. In addition, a PSC with an OA-coated PbSO_4_(PbO)_4_ quantum dot maintained an initial efficiency of 90% after 280 h of operation [122]. Quantum dots improved the energy level matching at the interface between the HTL and the perovskite layer, thereby ultimately reducing the loss of holes at the interface. This is conducive to the extraction of holes as well as the improvement in the FF, short-circuit current, and conversion efficiency of the PSCs. An in situ interface defect contact passivation strategy was developed where PbS quantum dots are used as passivation agents to reduce traps in perovskite films. The presence of PbS facilitated the crystallization of perovskite, passivated the interface and grain boundary defects, and reduced the nonradiative recombination; hence, the hysteresis effect and charge collection of the device improved significantly. In terms of the overall device performance, the PbS-optimized PSCs demonstrated a maximum efficiency of 21.07%, which was significantly higher than that of a comparative device (19.35%). [124]. The introduction of quantum dots effectively promoted the extraction of interface charges and inhibit charge recombination, thereby increasing the efficiency of the device.

#### 4.2.2. Carbon Material

Carbon materials offer advantages of low price, high conductivity, hydrophobicity, and chemical stability [139,140,141,142]. The introduction of carbon materials into PSCs can effectively reduce the cost of the PSCs and improve the efficiency and stability of the device [143,144,145,146,147]. Carbon materials of different dimensions include 0D fullerenes and their derivatives, 1D carbon nanotubes, carbon fibers, 2D graphene, graphyne, and 3D graphite, all of which have been used in PSCs [148,149,150,151].

The applications of 1D carbon materials in PSCs were mainly carbon nanotubes, which were used as additives for the hole transport layer due to their good hole transport properties. In 2016, Snaith’s research group used P3HT to wrap single-walled carbon nanotubes to form a supramolecular nanocomposite and then filled it with insulating PMMA as the HTL. One-dimensional carbon nanotubes are more effective in improving the hole-transport rate. It was reported that they improve the thermal stability and moisture resistance of the composite HTL of the device and achieve a maximum PCE of 15.3% [144]. Graphene-based materials modify the ETL/perovskite interface, thereby improving the extraction and transmission capabilities of the ETL for photogenerated electrons. The series resistance of the entire device is reduced, thereby effectively improving the PSC performance. Furthermore, extraction by blending fullerene into an anti-solvent is an effective method to passivate the perovskite layer.

Carbon materials with 2D structure had unique properties such as large specific surface area, better carrier fluidity, high thermal conductivity, and high light transmittance. In a recent study, Duan et al. developed a chemical stitching strategy to develop a scalable graphene oxide nanosheet (BJ-GO) sandwiched between a non-carbamate perovskite and a no-dopant HTL. In this configuration, iodide ions are physically fixed by a dense 2D network, and lead defects are chemically passivated by coordination bonds. In addition, BJ-GO with an adjustable surface energy increases the carrier mobility of the highly ordered HTL by an order of magnitude. Finally, the 35.80 cm^2^ PSC module using this heterostructure indicates an authentication efficiency of 15.3%. The packaged PSC module retained more than 91% of its initial efficiency after 1000 h of damp and heat testing at 85 °C under 85% relative humidity [130]. Wang et al. formed strong Pb–Cl and Pb–O bonds between an FA_x_MA_1−x_Pb_1+y_I_3_ film with a Pb-rich surface and a chlorinated graphene oxide layer. This structure enabled the selective extraction of photogenerated charge carriers and prevented the loss of decomposing components in the soft perovskite, thereby reducing damage to the organic charge transport semiconductor. When tested at AM1.5G sunlight and at the MPP for 1000 h at 60 °C, the PSCs with an active area of 1.02 cm^2^ maintained 90% of their initial efficiency (the initial value was 21%) [131]. These studies indicated that the application of carbon materials in PSCs will be an important part of realizing low-cost commercialization and large-scale manufacturing of PSCs.

#### 4.2.3. Oxide

Owing to their excellent chemical stability, high electron mobility, and an energy level structure that matches that of perovskites, oxide materials are considered to be an ideal option for ensuring the stability of perovskite surfaces [118,135,136,152]. The stability of the oxide was quite excellent. A protection strategy for in-situ generation of passivation layer was proposed by Huang et al. [132]. A perovskite reacts with sulfate or phosphate ions to form a thin and dense inorganic oxygen-containing lead salt layer on its surface. Subsequently, the oxygen-containing lead salt layer on the surface forms a strong chemical bond with the perovskite and provides better protection against harmful stimuli under atmospheric and light conditions. The lead oxide layer increases the carrier recombination life and improves the efficiency of the solar cell by 21.1%. Under AM 1.5G irradiation, the packaged device with a stable lead oxide layer demonstrates a stable output at 65 °C for 1200 h, maintaining 96.8% of its initial efficiency [132]. Defect states caused the inherent instability of hybrid perovskite materials. Bai et al. wrapped perovskite particles in the core–shell geometry of an oligomeric SiO_2_ (OS) matrix. This can passivate the defects at the surface and grain boundaries as well as stabilize the nanoscale grains. The trap density in the OS-coated perovskite decreases significantly, and the carrier lifetime is prolonged. The efficiency of the PSC increases by 21.5%; in addition, a high open-circuit voltage of 1.15 V and an FF of 81% were achieved. Under a full-spectrum of irradiation, the device maintains an initial efficiency of 80% after aging for more than 5200 h [133].

Inorganic nickel oxide was considered to be an excellent hole transport material due to its excellent photoelectric properties and extremely high stability, and was widely used in inverted structure PSCs. It was also used as an interface modification material for normal structure PSCs. A simple method for modifying perovskite films using NiO nanocrystals were reported. This method improves the interface contact characteristics, enhances the charge transport kinetics, and inhibits a charge recombination in a fabricated device, thereby further increasing *V_oc_* and *J_sc_*. The optimal open-circuit voltage of the PSCs was 1.15 V, the short-circuit current density was 23.89 mA/cm^2^, and the FF was 70.95%. Furthermore, the energy conversion efficiency increased from 16.31% without treatment to 19.47% after surface modification [134]. As an interface layer, nickel oxide provided a strong surface passivation effect and a suitable energy level arrangement, thereby significantly improving the photoelectric performance at the interface. Xiang et al. proposed a method for forming a nickel oxide intermediate layer between a perovskite layer and an HTL at low temperatures. Compared with inorganic PSCs prepared without a nickel oxide layer, the fabricated device demonstrated excellent efficiency. The open-circuit voltage of the device increased by 100 mV, and a PCE of 13.6% was obtained [135]. Li et al. also successfully prepared a NiO_x_ nanomicelle solution and applied it to *n*–*i*–*p* planar PSCs. The device exhibited a maximum efficiency of 21%, its *V_oc_* increased from 1.10 to 1.14 V, and its reproducibility was high. Under 1 sun intensity, the stability of the NiO_x_ device increased by four times, and its initial efficiency of 90% was maintained for 1200 h [136]. These studies shows that compared with the pure Spiro device, the device with a nickel oxide layer improves the extraction capacity of the hole carriers and exhibits an excellent operational stability.

## 5. Conclusions and Outlook

Passivation at the interface is essential for obtaining low-defect perovskite films with high-crystallinity. Furthermore, by changing the energy level to enhance the charge and hole transport capacity of the perovskite layer, high-efficiency PSCs can be obtained. Passivation at the interface can increase the grain size of the perovskite film, facilitate the formation of pure phases, reduce surface pores, and yield a smooth and dense perovskite layer. The inter-interface contact reduces the formation of interfacial defects and recombination of carriers, extends the moving distance of the excitons, and facilitates the transport of interfacial carriers, thereby allowing efficient and stable PSCs to be prepared. The modification and regulation of the interface in the device can effectively suppress the recombination rate of carriers at the interface and improve the PCE of the device. An analysis of the interface passivation materials will enable researchers to identify suitable passivation materials, understand the energy loss mechanism of the device interface, design a better interface for perovskite devices, and further improve the efficiency of PSCs.

Although great progress has been made in interface engineering to achieve efficient and stable PSCs, there is still a great gap in commercial applications. Generally, potential difficulties and challenges are existing in the widespread application of PSCs, i.e., large-area manufacturing process, long-term stability, device packaging technology, and the toxicity of perovskite film. These difficulties and challenges have seriously hindered the development of the industrialization of PSCs, however, its vigorous development and wide application are an irresistible trend of the times. With the development of science and technology, it is believed that many existing staple problems in PSCs will be well resolved.

## Figures and Tables

**Figure 1 nanomaterials-11-00775-f001:**
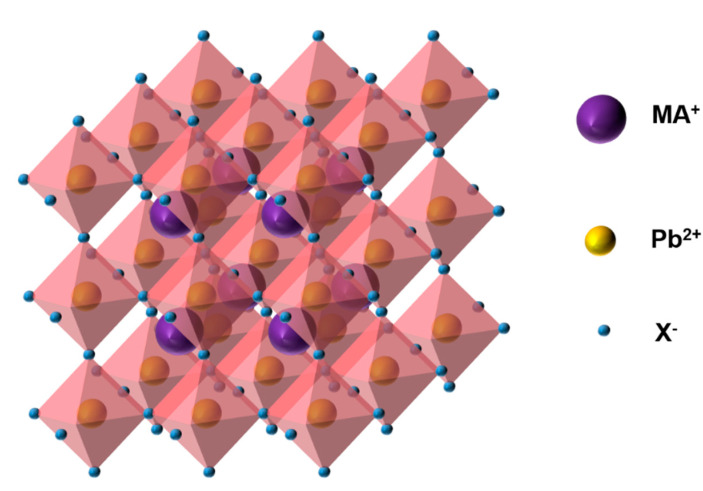
Structure of MAPbX_3._

**Figure 2 nanomaterials-11-00775-f002:**
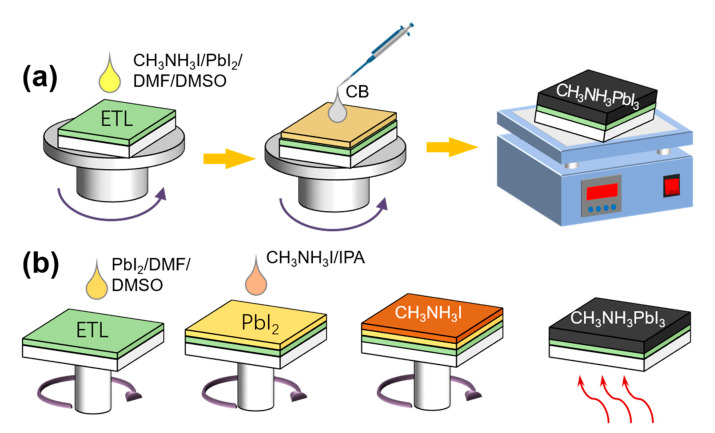
(**a**) One- and (**b**) two-step methods.

**Figure 3 nanomaterials-11-00775-f003:**
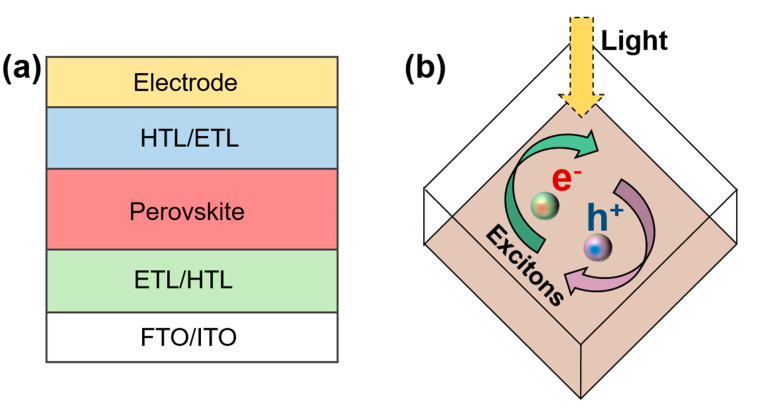
(**a**) Structure of PSC and (**b**) schematic diagram of exciton separation.

**Figure 4 nanomaterials-11-00775-f004:**
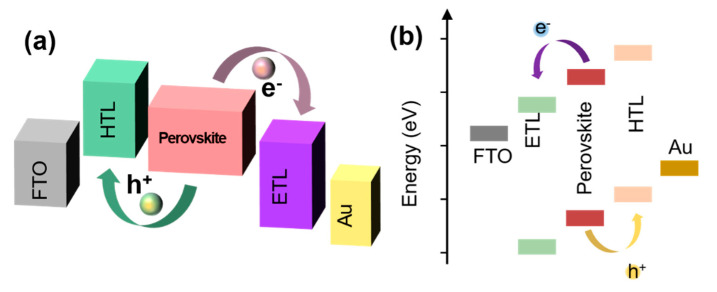
Schematic diagram of charge transfer of (**a**) inverted and (**b**) conventional PSCs.

**Table 2 nanomaterials-11-00775-t002:** Performance summary of chemical passivation PSCs.

Passivation Materials	Perovskite	*J_sc_* (mA·cm^−2^)	*V_oc_* (V)	*FF* (%)	*PCE* (%)	Year	Ref.
PA	Cs_0.05_FA_0.81_ MA_0.14_PbI_2.55_Br_0.45_	22.39	1.10	79.6	19.6	2019	[61]
PEA	Cs_0.05_FA_0.81_ MA_0.14_PbI_2.55_Br_0.45_	22.19	1.12	80.3	19.2	2019	[61]
VA	Cs_0.05_FA_0.81_ MA_0.14_PbI_2.55_Br_0.45_	22.42	1.12	80.1	20.1	2019	[61]
PAA	Cs_0.05_FA_0.81_ MA_0.14_PbI_2.55_Br_0.45_	22.62	1.15	79.0	20.6	2019	[61]
D4TBP	Cs_0.05_FA_0.81_ MA_0.14_PbI_2.55_Br_0.45_	22.51	1.16	78.5	21.0	2019	[61]
PA+PEA	Cs_0.05_FA_0.81_ MA_0.14_PbI_2.55_Br_0.45_	22.54	1.15	78.5	20.4	2019	[61]
CS0	FA_0.8_MA_0.2_PbI_3_	24.07	1.07	81.7	21.04	2020	[98]
CS1	FA_0.8_MA_0.2_PbI_3_	24.14	1.10	84.2	22.37	2020	[98]
CS2	FA_0.8_MA_0.2_PbI_3_	23.74	1.12	83.6	22.17	2020	[98]
TBPO	CsFAMAPb(IBr)_3_	24.10	1.14	81.0	21.9	2020	[109]
TPPO	CsFAMAPb(IBr)_3_	23.91	1.12	80.1	21.9	2020	[109]
DPP-DTT	CsPbI_2_Br	15.02	1.29	77.86	15.14	2019	[110]
PMMA	Cs_0.05_(FA_0.85_MA_0.15_)_0.95_Pb(I_0.85_Br_0.15_)_3_	21.55	1.16	69.40	17.43	2020	[111]
PCBM	Cs_0.05_(FA_0.85_MA_0.15_)_0.95_Pb(I_0.85_Br_0.15_)_3_	22.34	1.15	69.56	17.80	2020	[111]
PMMA:PCBM (1:2)	Cs_0.05_(FA_0.85_MA_0.15_)_0.95_Pb(I_0.85_Br_0.15_)_3_	22.73	1.17	69.87	18.63	2020	[111]
PMMA	MAPbBr_3_	1.41	6.63	67.7	6.32	2020	[112]
Py	MAPbI_3_	22.64	1.12	75.02	19.02	2019	[113]
PTT	MAPbI_3_	22.52	1.10	75.63	18.74	2019	[113]
2-MP	MAPbI_3_	22.61	1.16	77.44	20.28	2019	[113]

**Table 3 nanomaterials-11-00775-t003:** Performance summary of inorganic materials passivation PSCs.

Passivation Materials	Perovskite	*J_sc_* (mA·cm^−2^)	*V_oc_* (V)	*FF* (%)	*PCE* (%)	Year	Ref.
PbSO_4_(PbO)_4_	CH_3_NH_3_PbI_3_	24.68	1.10	75.0	20.35	2020	[122]
MAPbBr_0.9_I_2.1_	CH_3_NH_3_PbI_3_	19.51	0.948	72.0	13.32	2016	[123]
PbS	Cs_0.05_(FA_0.85_MA_0.15_)_0.95_ Pb(I_0.85_Br_0.15_)_3_	23.06	1.146	79.82	21.07	2020	[124]
CsPbBrCl_2_	CH_3_NH_3_PbI_3_	23.40	1.15	80.0	21.15	2019	[125]
CsPbBr_1.85_I_1.15_	CsFAMAPb(I_0.85_Br_0.15_)_3_	23.35	1.14	79.0	21.14	2019	[126]
SnO_2_	CsMAFAPbI_3_Br_3–x_	20.70	1.11	75.0	17.30	2019	[127]
CdTe	CH_3_NH_3_PbI_3_	22.42	1.10	78.0	19.19	2018	[128]
CsPbCl3:Mn	CH_3_NH_3_PbI_3_	22.03	1.11	76.0	18.57	2017	[129]
TEOS-GO	CsFAPbI_3_	24.02	1.06	78.24	19.92	2020	[130]
MPTES-GO	CsFAPbI_3_	23.90	1.10	77.69	20.42	2020	[130]
APTES-GO	CsFAPbI_3_	24.12	1.12	80.51	21.75	2020	[130]
GO	[CH(NH_2_)_2_]_x_ [CH_3_NH_3_]_1−x_Pb_1+y_I_3_	23.86	1.09	78.0	20.29	2019	[131]
Cl-GO	[CH(NH_2_)_2_]_x_ [CH_3_NH_3_]_1−x_Pb_1+y_I_3_	23.82	1.12	79.0	21.08	2019	[131]
PbSO_4_	Cs_0.05_FA_0.81_MA_0.14_PbI_2.55_Br_0.45_	22.63	1.16	80.4	21.11	2019	[132]
OS	MAPbI3	22.70	1.15	80.9	21.10	2019	[133]
OS	FA_0.85_MA_0.15_Pb(I_0.85_Br_0.15_)_3_	23.10	1.15	81.10	21.50	2019	[133]
NiO nanocrystals	MAPb (IBr)_3_	23.89	1.15	70.95	19.47	2018	[134]
NiO_x_	CsPbI_2_Br	14.26	1.25	76.0	13.60	2020	[135]
NiO_x_	CsFAMAPb(IBr)_3_	23.82	1.14	79.80	21.66	2020	[136]

## Data Availability

The data that support the findings of this study are available from the corresponding author upon reasonable request.

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
