# Peer review of "Review of Interface Passivation of Perovskite Layer"

_nanomaterials, 2021, doi:10.3390/nano11030775_

Round 1

Author Response

REVIEWER REPORT(S): 
Referee: 1 
Recommendation:  Reconsider - after major revisions 

Comments to the Author 
The manuscript reports an overview of interface passivation strategies in perovskite solar cells. The authors refer to the benefits of perovskite solar cells, the solar cell architecture and focus on the strategies that enhance the efficiency of perovskite devices through passivation of the interlayer interfaces. The study is interesting, however, there are many issues that must be addressed especially in the first two chapters which must be revised extensively to make scientific and chemically sense. I recommend publishing at Nanomaterials after major revisions.

Answers: Thank you for your highly appraisal of our work and valuable comments.

Additional comments to the authors:

  1. Line 30, the limit currently is 25.5%.

Answers: Thank you for your comments.

 We have made corrections in the paper. Current single-junction PSCs have achieved a certified power conversion efficiency (PCE) of 25.5%.

  1. Line 33, the perovskite materials are not organic, they are hybrid (both organic and inorganic).

Answers: Thank you for your comments.

We have made corrections in the paper. In PSCs, the most typically used perovskite material is the organic–inorganic metal halide perovskite, CH3NH3PbX3 (MAPbX3).

  1. Line 35, MAPbI3 does not have short carrier diffusion length.

Answers: Thank you for your comments.

We have made corrections in the paper. The structure of MAPbX3 is shown in Figure 1. Its disadvantages include poor stability and low open-circuit voltage.

  1. I recommend the authors to study these reviews to better understand the structure and properties of perovskite materials and revise their introduction accordingly. (Chem. Rev. 2016, 116, 12956−13008, J. Mater. Chem. A, 2020, 8, 21356-21386, https://doi.org/10.1002/aenm.201700264, DOI: 10.1038/NPHOTON.2014.134.)

Answers: Thank you for your comments. We have read these articles and modified the structure and properties of perovskite materials. The proposed literature has been cited in the article.

  1. Please include more figures in the manuscript.

Answers: Thank you for your comments. We have made further improvements. Figures 4, 5, and Tables 1, 2, 3 were added to the manuscript.

  1. Line 291, the meaning of the sentence is unclear.

Answers: Thank you for your comments.

We have made further improvements. A small number of ammonium salts participate in the ion exchange reaction, and 2D perovskite is formed on the surface of the 3D perovskite film

  1. Line 358, there are no free charges on the surface of the films. Everything is charged balanced.

Answers: Thank you for your comments.

We have made further improvements. This sentence was deleted in the manuscript.

  1. Line 361, the meaning of the sentence is unclear.

Answers: Thank you for your comments.

We have made further improvements. The molecular hydrophilic chemical group and perovskite molecule combine to form a coordination bond, which is conducive to the transport of carriers.

  1. Please pay more attention and elaborate in detail the mechanism of each strategy on the passivation of the different layers in the device (chapters 3 and 4).

Answers: Thank you for your comments. We have made further improvements.

We have made changes in the relevant parts

  1. Line 488 needs references.

Answers: Thank you for your comments. The related literature has been cited in the article.

  1. Line 527-529, the statement is not true.

Answers: Thank you for your comments.

We have made further improvements. Passivation at the interface can increase the grain size of the perovskite film, facilitate the formation of pure phases, reduce surface pores, and yield a smooth and dense perovskite layer; however, it can increase the thickness of the perovskite layer and the electron or hole transport layer.

Reviewer 2 Report

The authors have provided a very intersting review on passivation of hybrid perovskite layers for PSCs.

The work is well structured and complete despite of the enormous nuber of publication in the field, and I don't have scientific comments on it, here are my suggestions to improve the impact of the work:

(1) perovskite is very generic term, please consider modifying the title to: " a review of (surface and )interface passivation of hybrid perovskite layers", this is just a possibility of course!

(2) L27 "cynosural" I don't understand this term

(3) L33 MAPI perovskite is not anymore the most USED but it is certainly the most studied!

(4) I suggesst to integrate references on PSCs interface analysis and degradation upon solar exposure:

DOI:10.3390/ma12050726

DOI:10.1016/j.mtener.2018.04.005

DOI:10.1021/acsami.5b08038

(5) Figure 4 is confusing and not effective, I don't understand the meaning of it, I suggest to re-elaborate it in a more clear message

(6) The name of the section is too minimalistic, authors would need to read the whole text to find information, I suggest to be much more precise in the tiles of the sections to guide the reader through the manuscript: for example section 4.1.2 deals about the methods for the passivation of layered hybrid pérovskites to increase their stability, I thinck then that something like this should be in the section title.

(7) L287 I perfectly agree that the stability of PSCs versus light heat, atmosphere  and humidity are a key bottleneck, I suggest to include this in the abstract!

Regards

Author Response

REVIEWER REPORT(S): 
Referee: 2 
The authors have provided a very intersting review on passivation of hybrid perovskite layers for PSCs.

The work is well structured and complete despite of the enormous nuber of publication in the field, and I don't have scientific comments on it, here are my suggestions to improve the impact of the work:

Answers: Thank you for your highly appraisal of our work and valuable comments.

  1. perovskite is very generic term, please consider modifying the title to: " a review of (surface and )interface passivation of hybrid perovskite layers", this is just a possibility of course!

Answers: Thank you for your comments. Review of surface and interface passivation of hybrid perovskite layer.

  1. L27 "cynosural" I don't understand this term

Answers: Thank you for your comments.

We have made further improvements. Among them, perovskite solar cells (PSCs) are the most typically investigated application.

  1. L33 MAPI perovskite is not anymore the most USED but it is certainly the most studied!

Answers: Thank you for your comments. In PSCs, the most typically studiedperovskite material is the or-ganic-–inorganic metal halide perovskite.

  1. I suggesst to integrate references on PSCs interface analysis and degradation upon solar exposure: DOI:10.3390/ma12050726; DOI:10.1016/j.mtener.2018.04.005 ; DOI:10.1021/acsami.5b08038.

Answers: Thank you for your comments. The proposed literature has been cited in the manuscript.

  1. Figure 4 is confusing and not effective, I don't understand the meaning of it, I suggest to re-elaborate it in a more clear message.

Answers: Thank you for your comments. We made changes in Figure 4 to better understand.

Figure 4. Schematic diagram of charge transfer of a) inverted and b) conventional PSCs.

  1. The name of the section is too minimalistic, authors would need to read the whole text to find information, I suggest to be much more precise in the tiles of the sections to guide the reader through the manuscript: for example section 4.1.2 deals about the methods for the passivation of layered hybrid perovskites to increase their stability, I thinck then that something like this should be in the section title.

Answers: Thank you for your comments. For a better understanding, the corresponding title had been replaced.

  1. L287 I perfectly agree that the stability of PSCs versus light heat, atmosphere and humidity are a key bottleneck, I suggest to include this in the abstract!

Answers: Thank you for your comments. We have made further improvements in abstract.

Perovskite solar cells (PSCs) are the most promising substitute for silicon-based solar cells. However, their power conversion efficiency and stability must be improved. The stability of PSCs versus light heat, atmosphere and humidity are also a key bottleneck. The recombination probability of photogenerated carriers at each interface in PSCs is much greater than the recombination probability of their bulk phase. The interface of a perovskite polycrystalline film is typically considered to be a defect-rich area, which is the main factor that limits the efficiency of PSCs. This review introduces and summarizes practical interface engineering techniques for improving the efficiency and stability of organic–inorganic lead halide PSCs. First, the effect of defects at the interface of PSCs, energy level alignment, and chemical reactions on the efficiency of PSCs are summarized. Subsequently, the latest developments pertaining to the modification of perovskite layers with different materials are discussed. Finally, the prospect of efficient and long-term stable PSCs through interface engineering is presented.

Reviewer 3 Report

The authors attempt a review on the interface processes in perovskite solar cells. However it seems that they do not dedicate enough efforts to the writing of the manuscript, since many sentences are meaningless, even in the abstract and in the conclusions. Even the explanation of the basic photovoltaic effect (2.1 parahrapg) is misleading. Thereofore this review is unable to get a sensible message to the reader. Accordingly, in my opinion, the manuscript is not acceptable for publication in Nanomaterials.

Author Response

REVIEWER REPORT(S): 
Referee: 3 
The authors attempt a review on the interface processes in perovskite solar cells. However it seems that they do not dedicate enough efforts to the writing of the manuscript, since many sentences are meaningless, even in the abstract and in the conclusions. Even the explanation of the basic photovoltaic effect (2.1 parahrapg) is misleading. Thereofore this review is unable to get a sensible message to the reader. Accordingly, in my opinion, the manuscript is not acceptable for publication in Nanomaterials

Answers: Thank you for your valuable comments. We greatly appreciate your time on this manuscript. We have made further improvements. We have modified the theory of existence proposed in the manuscript. In order to increase the readability of the manuscript, we have made a lot of changes to the content and grammar of the manuscript and added pictures and tables. After addressing the issues raised and including our latest results, we feel the quality of the revised paper is much improved.

Eg 2.1

A PSC comprises a positive electrode layer, a hole/electron transport layer, a light absorption layer, an electron/hole transport layer, and a negative electrode layer (Figure 3a).[1, 13, 30, 31] The transport layer transports electrons and holes while preventing them from entering. Perovskite materials have a high dielectric constant and low excitation energy. After absorbing sunlight, the perovskite layer first absorbs photons to generate electron–hole pairs.[32, 33] Owing to the difference in the binding energy of excitons in perovskite materials, these carriers either become free carriers or form excitons. (Figure 3b) These perovskite materials tend to have a low carrier recombination probability and high carrier mobility; therefore, the carrier’s diffusion distance and lifetime are long. These uncomplexed electrons and holes are collected by the electron transport layer (ETL) and hole transport layer (HTL), respectively, i.e., the electrons are transported from the perovskite layer to the isoelectron transport layer and finally collected by the FTO. The holes are transported from the perovskite layer to the hole transport layer and are finally collected by the metal electrode. These processes are accompanied by some carrier losses, such as the reversible recombination of electrons and holes in the ETL and perovskite layer, respectively; the recombination of electrons and holes in the ETL and HTL, respectively; and the recombination of electrons and holes in the perovskite layer and HTL, respectively. The loss of these carriers should be minimized to improve the overall performance of the device. Finally, a photocurrent is generated through a circuit connecting the FTO and metal electrode.[31, 34, 35] One of the key methods for improving the PCE of PSCs is by improving the carrier separation efficiency of the materials.

Reviewer 4 Report

The authors discussed modification of Perovskite layers by different materials for making better solar cells. Perovskite-based solar cells have become the most potential substitute for silicon-based solar cells due to their high efficiency, low cost, and simple process which makes this manuscript an interesting review article for the researchers in this field.

I believe this manuscript should be published in its current format.

Author Response

REVIEWER REPORT(S): 
Referee: 4

The authors discussed modification of Perovskite layers by different materials for making better solar cells. Perovskite-based solar cells have become the most potential substitute for silicon-based solar cells due to their high efficiency, low cost, and simple process which makes this manuscript an interesting review article for the researchers in this field. 

I believe this manuscript should be published in its current format.

Answers: Thank you for your highly appraisal of our work.

Reviewer 5 Report

In this manuscript, the authors attempt to review the literature on the subject of interface passivation of perovskites in perovskite solar cells (PSCs). Although they have reviewed the literature on the field up to date to a good degree, the organization of the manuscript is poor and the description of the associated materials and methols lacks coherency and an in depth, comprehensive, discussion.

For example, the two main sections of the manuscript are distinguished by the type of the materials used for interface passivation (organic vs. inorganic) but, then, in the associated subsections of the first section they include a subsection entitled chemical passivation (where different materials adn methods are discussed) while all the other ones are entitled with the type of material discussed there in ! It is not also clear based on what criteria the selected materials were chosen in the review as there are many other inorganic and organic materials (such as, for example, organic small molecules, conjugated polymers, metal chalcogenides, graphene-based materials, inorganic salts) that have employed for defect passivation in PSCs. Moreover, some of the passivation materials evaluated are related to passivate surface and/or bulk defects of the perovskite (rather than interfacial defects) such as grain boundaries and other defects.

Furthermore, no tables or graphs are included to help the reader in evaluating the effectiveness of differerent passivation materials, methods or strategies employed in various PSCs in enhancing the device efficiency and stability. Finally, the outlook subsection is very poorly written as it does not provide the authors perespective on the challenges still present and some ideas in order to overcome them for PSCs to be able to reach higher performance.

Author Response

Answers: Thank you for your valuable comments. We have made further improvements in this manuscript. The opinions put forward are important guidance for us to revise the manuscript. We have modified the theory of existence proposed in the manuscript. In order to increase the readability of the manuscript, we have made a lot of changes to the content and grammar of the manuscript and added pictures and tables. After addressing the issues raised and including our latest results, we feel the quality of the revised paper is much improved.

Round 2

Reviewer 1 Report

all issues settled.

Author Response

Thank you for your valuable comments. We have made further improvements to the grammar in this manuscript. we feel the quality of the revised paper is much improved.

Reviewer 3 Report

The authors have strongly  revised the manuscript, according to the referees' suggestions, so that it can be accepted for publication in the present form.

Author Response

(The authors gave the same response as above.)

Reviewer 5 Report

The authors have taken into consideration the reviewers comments and certainly improved the quality and readibility of their manuscript. However, the review still fails to comment on some of the most widely used passivation layers used in perovskite solar cells as for example inorganic ones such as LiF or organic small molecules and the criteria of the selected materials chosen to be discussed have not clearly established. The review also continues ro lack coherency and focus, possibly in an effort to be as inclusive as possible.

Furhtemore, as I previously noted, a good review article not only includes an extensive summary of previous works but also provides the authors' own perspectives on the topics discussed. In my opinion, this still lacks in the current revision of the manuscript.

Therefore, my opinion is to reject the revised manuscript.

Author Response

REVIEWER REPORT(S): Referee: The authors have taken into consideration the reviewers comments and certainly improved the quality and readibility of their manuscript. However, the review still fails to comment on some of the most widely used passivation layers used in perovskite solar cells as for example inorganic ones such as LiF or organic small molecules and the criteria of the selected materials chosen to be discussed have not clearly established. The review also continues ro lack coherency and focus, possibly in an effort to be as inclusive as possible. Furhtemore, as I previously noted, a good review article not only includes an extensive summary of previous works but also provides the authors' own perspectives on the topics discussed. In my opinion, this still lacks in the current revision of the manuscript. Therefore, my opinion is to reject the revised manuscript. Answers: Thank you for your valuable comments. In the second and third parts of the manuscript, we have made a summary after reading a large number of articles. In the fourth part of the manuscript, the content is not coherent enough, and it really my point of view. Based on your suggestions, we read the narrative style of a good review and made a further improvement to the language and content of the manuscript. We have made major changes to the content of the fourth part. According to the suggestions, we added our own opinions and deleted some unimportant sentences. As for the classification of materials, as you mentioned, it is to discuss as many as possible. We have made some adjustments to it. In the continuous revision of this review, we have more in-depth thinking about how to write a review, which has allowed us to make great progress. Thanks again!

Round 3

Reviewer 5 Report

I would like to commend the authors for making in this round of review a considerably better effort not only to revise the manuscript according to the reviewers' comments but, more importantly, to comprehensively improve the manuscript's quality, presentation and organization as well as to give some personal perspectives for the challenges required to overcome in the interfacial passivation of perovskite solar cells. Therefore, I would be happy to recommend that the revised manuscipt can be accepted without further changes.